# OpenReview forum: "An Adversarial Perspective on Machine Unlearning for AI Safety"
_TMLR — Accepted by TMLR_

### Review · Reviewer_P7SZ · 2024-12-28

**Summary Of Contributions:**

The paper studies whether unlearning methods are truly scrubbing knowledge from the model's weights or only performing robust refusal. It proposes five interventions to evaluate this question, ranging from jailbreak-based attacks, interpretability techniques like steering / neuron pruning, and minimal finetuning with a small number of examples. These interventions are enough to recover dual-use knowledge that RMU claims to unlearn.

**Audience:**

Yes

**Broader Impact Concerns:**

None, barring the potential harms of over claiming (as described in weaknesses).

**Claims And Evidence:**

Yes

**Requested Changes:**

**Major**

1. Add more unlearning methods to make any claims about ‘unlearning’ in general rather than just RMU, such as those studied in [2].
2. Remove claims that ‘black box unlearning is insufficient’ without theory / experiments to support this.
3. Add better discussion of related work, as mentioned in W3.
4. The main paper has few actual results / figures / tables. It would be nice to shift salient experimental results from the appendix to the main paper, and also include a discussion of what's there in the appendix in the main paper.

**Minor**

1. WMDP not WDMP in page 2
2. Definition of retain set is wrong in experimental setup. Retain set can include any knowledge to retain, and not necessarily neighbouring samples
3. Sec 4 appendix references give the impression that the referenced appendix has the result, but it only has the setup. Please make this explicit. Eg: “we provide a comprehensive evaluation of how different configurations affect the pace of hazardous knowledge recovery. For further details see Appendix E.1” could be ‘further setup details are provided in E.1”

**References**

[1] Tamirisa, Rishub, et al. "Tamper-resistant safeguards for open-weight llms, 2024." *URL https://arxiv. org/abs/2408.00761*.

[2] Che, Zora, et al. "Model Manipulation Attacks Enable More Rigorous Evaluations of LLM Capabilities." *Neurips Safe Generative AI Workshop 2024*.

[3] Doshi, Jai, and Asa Cooper Stickland. "Does Unlearning Truly Unlearn? A Black Box Evaluation of LLM Unlearning Methods." *arXiv preprint arXiv:2411.12103* (2024).

[4] Goel, Shashwat, et al. "Towards adversarial evaluations for inexact machine unlearning." *arXiv preprint arXiv:2201.06640*(2022).

[5] Sharma, Aakash Sen, et al. "Unlearning or concealment? a critical analysis and evaluation metrics for unlearning in diffusion models." *arXiv preprint arXiv:2409.05668* (2024).

**Strengths And Weaknesses:**

**Strengths**

1. Shows RMU does not lead to robust weight-space scrubbing of harmful knowledge using many different interventions
2. Makes an important point that unlearning methods should do more than just robust refusal for them to get separate attention

**Weaknesses**

1. Only benchmarks RMU, generalising observations on RMU to ‘unlearning methods’ when claiming unlearning methods behave like refusals. The RMU method is weak, merely adding noise to early layer representations, which is not necessarily true for all unlearning methods. This paper could be used to claim “unlearning merely performs robust refusal” without sufficient evidence in future discourse. Further, the paper claims RMU is state of the art for LLM unlearning, which it is not. Improved methods like TAR [1] were released 5 months ago. Please refrain from using the term ‘state of the art’ in this paper, or include actual SOTA methods.
2. The claim that “black-box evaluations are inadequate for unlearning” is weakly supported. The paper shows some white-box evaluations can recover unlearnt knowledge. However, it makes no attempt to show black-box evaluations cannot do this. In fact, [3] could recover the knowledge RMU claims to unlearn with few-shot prompting (a black-box eval). Further, WMDP evaluated using probing (a white-box eval) and still couldn't recover the knowledge. I think it's more important to have adversarial adaptive evaluations rather than white-box, and this claim could be wrong. It is possible that black-box evaluations can expose poor unlearning as effectively as white-box in the limit, though it might be harder to design such black-box evals.
3. The related work section is far from adequate. (1) it would be useful to cite (likely) parallel work like [2, 3] which showed similar results and compare contributions to it. (2) it misses citations and discussion of papers in unlearning outside LLMs which offer similar arguments (for eg [4], [5]). Related work should say ‘LLM Unlearning’ as many claims are untrue for the broader unlearning literature.

---

> ### Author Response · Authors · 2025-02-17
>
> We thank the reviewer for their time and comments. In this response, we detail the changes we will make to the paper and clarify some of the weaknesses raised.
>
> > Only benchmarks RMU
>
> In our paper we evaluate 3 methods: RMU, NPO and DPO. The first two are unlearning methods, whereas the last one is a standard safety training technique, which we treat as a ‘control group’. We chose RMU and NPO for five reasons:
> 1. They represent two major and different approaches: representation engineering and finetuning.
> 2. They are inherently deep unlearning methods. By this we mean that these methods actually attempt to remove target knowledge. For this reason, we discarded ECO [R6], which does not change the model and only adds guardrails. Similarly, we discarded TAR [R7], which only trains an already unlearned model to make it more robust to adversarial finetuning. Moreover, in the original evaluation, TAR is reported to be vulnerable to some instances of LoRA fine-tuning and is thus not adversarially robust.
> 3. Their performance is strong enough to be used in real-world applications. We discard theoretical methods that cannot be applied at scale such as LLMU  [R1], SCRUB [R2], SSD [R3], which either weren’t able to unlearn a meaningful amount of hazardous knowledge or their application resulted in drastic loss in utility [R4].
> 4. We do not use unlearning methods designed to unlearn specific knowledge (e.g.  the technique introduced in ‘Who is Harry Potter?’ [R5]) as in safety applications we want to unlearn fuzzy and broad topics.
> 5. We only include methods that claim to remove knowledge from model weights for a fair assessment. For this reason, we discard other state-of-the-art safety methods such as circuit breaking [R8], as they are not designed to remove knowledge and our statements could be misleading.
>
> In conclusion we chose the only 2 state-of-the-art unlearning methods that can be applied to broad safety topics.
>
> > The claim that “black-box evaluations are inadequate for unlearning” is weakly supported.
>
> Original black-box evaluations based on outputs suggest that unlearning methods completely remove information from a topic (~25% accuracy in 4 choices questions). We claim this is inadequate since the goal of unlearning is **not** to prevent outputs (as refusal training does) but to **remove information from model weights**. We thus believe **evaluations should take into account model weights to correctly assess if information is still encoded there**, making black-box evaluations inadequate. Although there might be some black-box methods that succeed (like few-shot as the reviewer mentioned), white-box will always provide a strictly stronger evaluation regime for unlearning.
>
> Concerning the results in [3], please note that in the case of RMU and WMDP Bio the black-box methods such as few-shot prompting were able to recover at most 39.1% accuracy (original accuracy: 66.5%). With our white box evaluations we were able to recover full original accuracy. _We will_ clarify this in the paper and make a more nuanced claim on what we mean by black-box here.
>
> Summary of requested changes:
>
> > Include additional methods like those in [2]
>
> As mentioned above, most of the methods presented in [2] are not framed as unlearning by the authors. We believe it is thus unfair to use them to support our claims since the authors did not try to claim their method **removes** information from model weights in the first place.
>
> > Remove claims that ‘black box unlearning is insufficient’ without theory / experiments to support this.
>
> As mentioned above, we will clarify our claims. We still believe that white-box evaluations are needed to assess if information is encoded in model weights robustly. We are happy to discuss this further.
>
> > Add better discussion of related work, as mentioned in W3.
>
> We will include the requested citations. Both concurrent works on unlearning robustness were released after our submission. We will include them together with “Do Unlearning Methods Remove Information from Language Model Weights?” as concurrent work.
>
> > Include more results in main paper
>
> We will try to improve the cross-referencing. Since we do not want to clutter the main body too much, is there any specific results the reviewer thinks should definitely be included in the main body?
>
> > Minor changes
>
> We will fix all minor issues raised.

---

> > ### Comment · Reviewer_P7SZ · 2025-03-04
> > **Concerns Adequately Resolved**
> >
> > The response and updates to the draft have resolved my main concerns adequately. Thanks!

---

> ### Author Response · Authors · 2025-02-17
>
> References for the comment above
>
> [R1] Yuanshun Yao, Xiaojun Xu, and Yang Liu. Large language model unlearning. arXiv preprint arXiv:2310.10683, 2023b.
>
> [R2] Meghdad Kurmanji, Peter Triantafillou, and Eleni Triantafillou. Towards unbounded machine unlearning. NeurIPS, 2023.
>
> [R3] Jack Foster, Stefan Schoepf, and Alexandra Brintrup. Fast machine unlearning without retraining through selective synaptic dampening. AAAI, 2024.
>
> [R4] Li, Nathaniel, et al. "The wmdp benchmark: Measuring and reducing malicious use with unlearning." arXiv preprint arXiv:2403.03218 (2024).
>
> [R5] Eldan, Ronen, and Mark Russinovich. "Who's Harry Potter? Approximate Unlearning in LLMs." arXiv preprint arXiv:2310.02238 (2023).
>
> [R6] Liu, Chris Yuhao, et al. "Large Language Model Unlearning via Embedding-Corrupted Prompts." arXiv preprint arXiv:2406.07933 (2024).
>
> [R7] Tamirisa, Rishub, et al. "Tamper-resistant safeguards for open-weight llms." arXiv preprint arXiv:2408.00761 (2024).
> [R8] Zou, Andy, et al. "Improving Alignment and Robustness with Short Circuiting." arXiv preprint arXiv:2406.04313 (2024).

---

### Review · Reviewer_puFF · 2025-02-10

**Summary Of Contributions:**

This work conducts a white-box study on the effect of unlearning methods. Its contribution is twofold. First, the findings of previous unlearning methods fail to erase the target knowledge (to be forgotten) from the model parameters. Second, it presents diverse approaches to recovering the target knowledge: fine-tuning, modifying the activations, and prompt attacking.

**Audience:**

Yes

**Broader Impact Concerns:**

I believe that this submission does not raise concerns about Broader Impact.

**Claims And Evidence:**

Yes

**Requested Changes:**

- Please resolve my concerns mentioned above.

## Minor
- Figures in the Appendix are not ordered by their indices, making it challenging to locate them.

**Strengths And Weaknesses:**

# Strengths
- This work conducts a white-box study on the model unlearning problem, providing rich intuition of the effect of unlearning methods.
- This work evaluates multiple recovery methods, showing diverse approaches to recovering hazardous capabilities.
- Multiple unlearning methods are evaluated to show the consistent effect of recovery methods.
- I appreciate the authors providing details in the Appendix, and planning to release the code.
- This paper is well-written. My reading experience was fluent.

# Weaknesses
- My main concern is the correlation between "recovered accuracy" and "recovering hazardous knowledge". The recovered accuracy might be a result of generalization. Because the "forget data" and the "retain data" are sampled from the same i.i.d. distribution. Hence the improvement of finetuning  (Section 4.1) on the "retain data" generalizes to the "forget data" and thus increases its accuracy on the "forget data". This fact makes the source of "recovered accuracy on the forgot data" vague (either from generalization or knowledge recovered?).

In conclusion, the results of Section 4.1 seem not to support the claim of "recovering hazardous knowledge", if without further discussion.

In Figure 2, we see that finetuning on Wikipedia fails to "recover" the accuracy of the RMU unlearned model, but on retain set recovers the accuracy. This might be a sign of generalizing to the same distribution because the retain set has the same distribution as the forget set.


- The proposed recovery methods access different resources: the "forget dataset" only, the original model only, or neither. While methods accessing different resources recover the hazardous capability, discussing the differences between these resource accessing makes the experiment better organized.

---

> ### Author Response · Authors · 2025-02-17
>
> We thank the reviewer for their time and comments. In this response, we detail the changes we will make to the paper and clarify some of the weaknesses raised.
>
> > correlation between "recovered accuracy" and "recovering hazardous knowledge"
>
> Thank you for bringing this question up. Evaluating these open-ended problems is a challenging problem for the community [R1]. If we understand correctly, the reviewer is concerned about the fact that the model might be recovering capabilities other than hazardous knowledge in the process. Regarding their first point
>
> >  Because the "forget data" and the "retain data" are sampled from the same i.i.d. distribution.
>
> we would like to highlight that retain dataset and forget dataset do not come from the same distribution. The authors of WMDP ensured that the former contains general biological knowledge and the later contains advanced knowledge from e.g. virology. So while the two datasets clearly concern the same domain and hence are doomed to have some mutual information there is a clear semantic difference.
>
> Since the mutual information is very small (or none) by design we discard the possibility of relearning information relevant for the forget dataset. We believe that recovering accuracy from 25% to ~65% at such high rates most likely indicates our ability to recover information that was never removed from model weights. This is further demonstrated by our methods that only rely on input perturbations (e.g. enhanced GCG) which cannot introduce new information in the model weights.
>
> > Results on Wikipedia dataset
>
> We would like to share some additional results from pilot experiments to provide better context on the results concerning wikipedia results. Throughout the project we tried multiple prompt designs and datasets for finetuning. For example, original version of Figure 2 (see below) reported results on the multiple choice version of wikipedia. One can see that the accuracy on WMDP-Bio (almost) monotonically increases with the number of samples. We decided to report results on wikipedia corpus instead so that it is consistent within each unlearning method (as WMDP-Bio retain is also comprised of pieces of text and not multiple-choice questions).
>
> See image here: https://postimg.cc/dhKgmvLm
>
> We have also tried finetuning on the cyber retain corpus, which is very distinct from Bio knowledge. There one can see that the original RMU model (which was unlearned on both bio and cyber datasets) immediately recovers an accuracy of 50% on Bio knowledge. Concerned about entanglement between unlearning of bio and cyber knowledge, we trained our own RMU models on bio dataset only and you can see that after 4000 samples model recovers accuracy of >55%.
>
> See image here: https://postimg.cc/KKW6TTL4
>
> These additional results show that finetuning attacks are rather delicate and small changes in prompt style or semantics (wikipedia vs cybersecurity) can heavily influence the results. Nevertheless, using datasets from different domains for finetuning still leads to knowledge recovery even if at lower pace.
>
> > The proposed recovery methods access different resources
>
> This is a good point. We will add a short note about various resources used by different methods and point the reader to the appendix where we will add a table with detailed comparison. Does the reviewer find it sufficient?
>
> > Minor: figure placement
>
> After closer inspection there was a rendering error which caused the figures to be displaced. We will fix it immediately so that the figure indices are correct.
>
> [R1] Rando, Javier, et al. "Adversarial ML Problems Are Getting Harder to Solve and to Evaluate." arXiv preprint arXiv:2502.02260 (2025).

---

### Review · Reviewer_otrc · 2025-02-11

**Summary Of Contributions:**

- The paper presents a thorough adversarial evaluation of unlearning methods, a crucial yet underexplored aspect of model safety.
- The comparison between unlearning and traditional safety fine-tuning is well-motivated, raising important questions about the efficacy of current approaches.
- The adaptive attacks (e.g., fine-tuning on unrelated examples, modifying activation space directions) provide novel insights into the vulnerabilities of state-of-the-art unlearning methods.
- The empirical results are compelling and suggest that existing unlearning techniques may not be as robust as previously assumed.

**Audience:**

Yes

**Claims And Evidence:**

Yes

**Requested Changes:**

- Extend the discussion on defenses, including potential improvements to existing unlearning techniques.
- Evaluate the attacks on additional benchmarks to for a complete comparison
- Provide more theoretical grounding for key claims, particularly regarding the equivalence of unlearning and safety fine-tuning.
- Analyze the trade-offs between robust unlearning and model utility in different safe contexts.

**Strengths And Weaknesses:**

Strengths:

- Comprehensive Adversarial Evaluation: The paper presents a rigorous white-box evaluation of state-of-the-art unlearning methods, exposing their vulnerabilities to adaptive attacks.
- Challenging the Assumptions of Unlearning: The analysis raises fundamental questions about whether unlearning truly removes hazardous knowledge or merely obfuscates it, contributing to a critical debate in model safety.
- The findings have significant implications for AI safety, emphasizing the need for more robust unlearning techniques to prevent adversarial recovery of hazardous knowledge.

Weaknesses:
- The paper has limited novelty - The findings on unlearning methods as well as the attacks used in this work have already been shown before by other papers on datasets other than WMDP
- Limited Discussion on Mitigations: While the paper effectively demonstrates vulnerabilities in unlearning, it lacks an in-depth exploration of how these issues could be addressed. Discussing potential defenses or improvements to existing unlearning methods would strengthen the work.
- Theoretical Underpinning: Some claims, particularly the argument that unlearning may be equivalent to safety fine-tuning in merely obfuscating knowledge, would benefit from additional theoretical support. Can a formal analysis clarify the difference between these mechanisms?
- Impact on General Model Utility: While the study focuses on recovering hazardous knowledge, it does not explore the trade-offs involved. For instance, does making unlearning more robust negatively impact the model's overall performance on safe tasks? Addressing this would provide a more holistic view of unlearning's effectiveness. Evaluation beyond MMLU on data that lies in the neighborhood should be useful

---

> ### Author Response · Authors · 2025-02-17
>
> We thank the reviewer for their time and comments. In this response, we detail the changes we will make to the paper and clarify some of the weaknesses raised.
>
> > The paper has limited novelty - The findings on unlearning methods have already been shown before by other papers on datasets other than WMDP
>
> As discussed in the paper, most existing works focused on unlearning narrow topics (e.g. Harry Potter [R4]). This work was the first, together with some concurrent findings, to evaluate unlearning for safety-relevant domains and methods such RMU and NPO.
>
> Furthermore, we have shown the importance of adaptive attacks, since for example, the original GCG evaluation in RMU [R1] did not recover performance. With our modifications, GCG was able to find universal adversarial strings that recover most of the model performance. Furthermore, we were the first ones to show the unreliability of LLM unlearning for safety so exhaustively, using a range of techniques from adversarial inputs, fine-tuning, and interpretability.
>
> > Limited Discussion on Mitigations
>
> Improving unlearning techniques is beyond the scope of this work. We demonstrate that existing unlearning methods are flawed and existing evaluations cannot capture whether information was truly removed. We hope these findings can improve rigor in future evaluations and inform better methods.
>
> > Theoretical Underpinning: Some claims, particularly the argument that unlearning may be equivalent to safety fine-tuning in merely obfuscating knowledge, would benefit from additional theoretical support.
>
> Providing theoretical support for these methods is extremely challenging as the community has not a good understanding of the mechanisms that underlie safety in LLMs. We, however, focus on an empirical validation of this claim. We can show that, under adversarial pressure, both safety fine-tuning and unlearning:
> 1. Leave models vulnerable to attacks.
> 2. This also implies that unlearning cannot remove information from model weights as initially believed and therefore only “obfuscates” knowledge, which is known to be a fundamental limitation of safety fine-tuning.
>
> > Impact of unlearning on general model utility.
>
> Evaluating the side effects of unlearning is outside the scope of our work. We focus on showing that unlearning does not provide reliable protections against adversaries as originally claimed. Also, the tradeoff between general utility and unlearning robustness was explored in [R3].
>
> ### Requested Changes
>
> > Extend the discussion on defenses.
>
> We are happy to highlight some available defenses and some learnings resulting from our experiments such as ensuring that there is no unlearning dimension present in the activations (possibly include a penalty term against that as a temporary measure). Furthermore, we can mention that adversarial evaluation using gcg might yield too optimistic results if the method is based on noising.
>
> > Evaluate the attacks on additional benchmarks to for a complete comparison.
>
> We evaluated our attacks on WMDP-BIO and WMDP-CYBER since they were used in the original RMU evaluation and which, to the best of our knowledge, are the only available benchmarks designed to test unlearning of safety-relevant topics. Note these benchmarks do not include harmful queries as common safety benchmarks do (e.g. how to build a bomb) but rather scientific questions that test knowledge that could later be used for harmful tasks (e.g. expertise in chemical reactions).
>
> > Provide more theoretical grounding for key claims, particularly regarding the equivalence of unlearning and safety fine-tuning.
>
> As mentioned above, finding theoretical grounds to compare methods is hardly possible due to both unlearning and safety fine-tuning being broad and ill-defined terms. We thus rely on empirical evidence of both families of methods behaving and failing in similar ways under adversarial pressure.
>
>
> [R1] Li, Nathaniel, et al. "The wmdp benchmark: Measuring and reducing malicious use with unlearning." arXiv preprint arXiv:2403.03218 (2024).
> [R2] Tamirisa, Rishub, et al. "Tamper-resistant safeguards for open-weight llms." arXiv preprint arXiv:2408.00761 (2024).
> [R3] Liu, Zheyuan, et al. "Towards safer large language models through machine unlearning." arXiv preprint arXiv:2402.10058 (2024).
> [R4] Lynch, Aengus, et al. "Eight methods to evaluate robust unlearning in llms." arXiv preprint arXiv:2402.16835 (2024).

---

### Author Response · Authors · 2025-02-17

We thank all the reviewers for their time and feedback. We have addressed all comments below and are working on an updated manuscript that we will update as soon as possible. Please, let us know if you have any additional feedback we can incorporate during this period.

---

### Author Response · Authors · 2025-02-26
**Paper updated**

Dear reviewers,

We have uploaded a new version of our work incorporating the feedback received. The most important changes are highlighted in blue, we summarize them next:

* Focusing the "Unlearning" related work paragraph on LLMs.
* Including a new paragraph commenting on concurrent work that happened after submission.
* Improved our justification for why black-box evaluations are insufficient for unlearning in the discussion.
* Included a new paragraph in the discussion commenting on possible mitigations and important lessons learned.
* Fixed some formatting problems and figures that were placed at the end of the document because of a rendering error.

We are happy to further incorporate feedback from the reviewers for camera-ready version if required.

---

### Decision · Action_Editor_NRyt · 2025-03-29

**Recommendation:** Accept as is

**Comment:**

This paper conducts a white-box study on the effect of unlearning methods. Its contribution is twofold. First, the findings of previous unlearning methods fail to erase the target knowledge (to be forgotten) from the model parameters. Second, it presents diverse approaches to recovering the target knowledge: fine-tuning, modifying the activations, and prompt attacking. For the strengths, this work conducts a white-box study on the model unlearning problem, providing rich intuition of the effect of unlearning methods. This work evaluates multiple recovery methods, showing diverse approaches to recovering hazardous capabilities. Multiple unlearning methods are evaluated to show the consistent effect of recovery methods. The authors address reviewers' concerns well in their rebuttal. Thus, I would like to recommend accept as is.

**Audience:**

Yes

**Claims And Evidence:**

Yes